# Moral Disengagement as a Self-Regulatory Cognitive Process of Transgressions: Psychometric Evidence of the Bandura Scale in Chilean Adolescents

**DOI:** 10.3390/ijerph191912249

**Published:** 2022-09-27

**Authors:** Andrés Concha-Salgado, Angélica Ramírez, Beatriz Pérez, Ricardo Pérez-Luco, Eduardo García-Cueto

**Affiliations:** 1Department of Psychology, Universidad de La Frontera, Temuco 4811322, Chile; 2School of Psychology, Universidad de Las Américas, Santiago 7500975, Chile; 3Department of Psychology, Universidad de Oviedo, 33003 Oviedo, Spain

**Keywords:** moral disengagement, scale, psychometric properties, violent antisocial behavior, delinquency, abusive behavior, prosocial behavior, empathy, adolescents, mediator

## Abstract

Moral disengagement is a process of cognitive restructuring that allows individuals to disassociate from their internal moral standards and behave unethically without feeling distressed. It has been described as a key predictor of maladaptive behaviors (e.g., delinquency, aggression, and cyberbullying) and as a mediator between individual variables and unethical outcomes (e.g., empathy and aggression). We aimed to provide evidence of validity based on the internal structure, reliability, and correlations with other constructs of the most used instrument to measure disengagement from moral self-sanctions: Bandura’s Mechanisms of Moral Disengagement Scale (MMDS). A non-probabilistic national sample of 528 Chilean adolescents from 14 to 18 years participated in the study. The results showed that the 10-item version of the MMDS had a unidimensional structure and good internal consistency. As expected, the MMDS-10 showed positive and medium correlations with abusive, violent antisocial, and delinquent behaviors and negative and medium associations with prosocial behavior and empathy. Additionally, moral disengagement fully mediated the relationship between empathy and violent antisocial behavior, supporting the hypothesis on moral disengagement as a self-regulatory cognitive process. The results confirm previous research, and the findings are discussed in terms of their implications for reducing the use of moral disengagement strategies in adolescence.

## 1. Introduction

Violence and criminal behavior in young people continue to be a growing threat. The cross-national study by van Dijk et al. [1], which analyzed crime patterns in 166 countries around the world, identified youth as one of the three determining indicators related to crime. Although we have observed a decrease in the contact of young people with the justice system in diverse cultural realities in recent years, such as in the US [2,3] or Chile [4], this problem is still relevant and requires intervention and improvements, primarily due to the consequences it has for those involved: young offenders or aggressors, victims, and their communities.

From a multidimensional ecological framework [5], the risk factors for the transgressions and violations of the law in youth are located at the individual, peer, family, community, and cultural levels. Related to individual factors, we highlight the self-regulation capacity of moral behavior and the mechanisms of moral disengagement (MD) from the contributions of Bandura’s Social Cognitive Theory [6,7], which are processes that affect not only the aggressor or perpetrator but also the victims [8] and bystanders of violence [9].

These moral processes are especially relevant in adolescence, since the tendency to justify one’s behavior develops from childhood [10], and also agency, namely the ability to self-motivate to achieve goals and reach desired futures through action plans (*forethought*), self-regulate one’s behavior through self-sanctions within an autonomous system contrasted with moral standards (*self-reactiveness*), and the metacognitive capacity to reflect on the efficacy and appropriateness of abilities, thoughts, and actions (*self-reflectiveness*) [11].

As people develop their moral identities, they adopt standards of right and wrong to guide their behavior [12]. However, being able to distinguish right from wrong does not mean that people behave accordingly [13]. The Social Cognitive Theory [6,11] identifies psychological mechanisms of self-regulation in interaction with the environment that are part of social learning and feed back the individuals’ behavior. In this theory, transgressiveness is regulated by two primary sources of sanctions—social sanctions and internalized self-sanctions—and both processes operate anticipatorily. People do things that give them satisfaction, and they refrain from behaviors that violate their moral norms and produce a sense of self-condemnation. When these self-sanctions are disconnected, MD arises.

### 1.1. Moral Disengagement

The Bandura Theory [6,14] analyzes MD as a process involving a series of interrelated cognitive mechanisms that legitimize the deactivation of the self-regulation processes from moral behavior. The MD is summarized as eight mechanisms: (1) *moral justification*, where harmful behavior is made acceptable by representing it in the service of moral values or moral purposes, (2) *euphemistic labeling*, where destructive behavior is made benign through an elaborate language presenting decontaminated and convoluted speech, (3) *advantageous comparison*, where one’s harmful behavior may seem benevolent compared with other people’s, (4) *displacement of responsibility*, where self-censorship reactions are avoided because people believe that they are not the real agent of their actions, (5) *diffusion of responsibility*, where responsibility can be diffused when a group engages in the same behavior, (6) *distortion of consequences*, where people can avoid facing the harm they cause or minimize it when one’s behavior is ignored, diminished, distorted, or disbelieved, (7) *dehumanization*, where self-censure reactions can be decoupled or attenuated by stripping people of human qualities, and (8) *attribution of blame*, where victims are blamed for bringing suffering to themselves, and self-exoneration is achieved by seeing the harmful behavior as forced by circumstances and not as a personal decision [12,14,15,16].

These mechanisms are grouped into four categories or *loci* of the self-regulation process, in which internal moral control can be detached from harmful behavior: cognitively reinterpreting immoral behavior (behavior *locus,* mechanisms 1, 2, and 3), diminishing personal responsibility (agency *locus*, mechanisms 4 and 5), ignoring or misrepresenting consequences (outcome *locus,* mechanism 6), and blaming or denigrating victims (victim *locus,* mechanisms 7 and 8) [11,14].

### 1.2. Moral Disengagement and Transgressive Behavior

Empirical studies have shown that MD is positively related to a wide range of deviant behaviors, such as criminal behavior [17,18], violent antisocial or aggressive behavior [7,19,20,21,22,23,24,25,26], substance use [27,28], or homophobic behavior [29].

Regarding violent behaviors, MD has been related to violence at the individual level in couples’ relationships [30,31,32] and, collectively, in contexts of protest [33]. Although the most abundant evidence indicates a positive effect of MD on both traditional bullying and cyberbullying [34,35,36,37,38,39,40,41], according to Gini et al. [42], significant correlations have been found in samples of young offenders and community samples, confirming that the MD mechanisms operate within a “normal” range of psychological functioning.

Several meta-analyses have contributed to clarifying the magnitude of the positive relationship between MD and deviant behavior. Férriz et al. [43] analyzed 20 articles on MD and crime and found a medium average effect size (*r* = 0.35). Wang et al. [44] reviewed 41 papers on MD and aggression and found an overall positive effect (*r* = 0.35). Gini et al. [42] synthesized 27 studies on the relationship between MD and aggression in children and young people aged 8–18 years, and they revealed that MD is positively associated with aggression with medium effect sizes (*r* = 0.27), bullying (*r* = 0.25), and cyberbullying (*r* = 0.31). Killer et al. [45] analyzed 47 independent samples on the relationship between MD and bullying roles in children and adolescents (7–19 years old), showing a positive relationship between MD and bullying (*r* = 0.31) and MD and victimization (*r* = 0.08) and a negative relationship between MD and defense (*r* = −0.11). Finally, Zhao and Yu [46] reviewed 38 studies on the relationship between MD and cyberbullying in adolescents and adults, concluding that they correlate positively and with a medium intensity (*r* = 0.341), also establishing age, gender, and cultural background (individualistic or collectivist) as moderators of the relationship.

As for the separate mechanisms, the majority use of unidimensional measures has limited the comparison of the differential predictive capacity of MD on antisocial behaviors [42], although in a recent longitudinal study, the mechanisms of moral justification, dehumanization, and distortion of consequences facilitated the transition from cyber gossip (comments, judgments, and spreading rumors about others) to cyberbullying [40]. In addition, Romera et al. [36] found that cognitive restructuring, which encompasses the mechanisms of moral justification, euphemistic language, and advantageous comparison, was the most relevant predictor of the perpetration of bullying and cyberbullying.

The evidence presented places MD as a risk factor for the onset, development, and persistence of deviant behaviors. It also highlights the importance of its study, even more so if the repetition of the MD process can gradually lead to the withdrawal of self-censure and thus the routinization of immoral behaviors [12,14]. From the previous review, we hypothesize that the MD will positively correlate with antisocial and violent behaviors.

### 1.3. Moral Disengagement, Prosocial Behavior, and Empathy

On the contrary, MD is negatively linked to prosocial and defensive behaviors toward the victims [7,13,16,47]. The classic study by Bandura et al. [16] concluded that adherence to self-sanctioning of offending behavior (low MD) is strengthened not only by a sense of empathy but also by taking personal responsibility for one’s actions and not minimizing their detrimental effects, which favors prosocial behavior. This link has also been observed in other areas, such as prosocial sports behaviors toward teammates and opponents [48,49].

Another variable related to MD is empathy, understood as the ability to be affected and share the emotional state of another, evaluate the reasons for the other’s state, and adopt their perspective [50]. Empirical studies determined that MD negatively correlates with empathy [23,25,26,47,51,52,53]. This relationship also occurs in cyberstanders or witnesses of cyberbullying who do not defend the victim [54]. This evidence allows us to hypothesize that MD will negatively correlate with prosocial behavior and empathy.

In the Latin American context, the links between MD and bullying [55], empathy and compassion [56], and the collective efficacy of schools in preventing bullying [57] have been studied in Mexico. In Venezuela, studies have been carried out on MD in ethical training within business education [58] and unethical behaviors in students and workers [59]. In Colombia, the link between MD and several variables, such as empathy and prosocial behavior in young lawbreakers [60], aggression and bullying in children and young people at risk [61], and in a simulated situation of coercion between peers to cheat academically [62,63] have been explored. Based on our literature review, no quantitative empirical studies on MD in Chile were found, so to our knowledge, this study would be the first.

### 1.4. Moral Disengagement as a Mediator

Bandura’s theory analyzes MD as a self-regulatory process [6]. The conceptualization of MD as a process is more consistent with a mediating role than a moderating role, typical of innate attributes or relatively stable traits. The process of MD decouples internal moral standards from one’s actions and facilitates engaging in immoral behavior, such as aggression [25,64,65].

Previous research has shown, both cross-sectionally and longitudinally, that MD works as a self-regulatory cognitive process, mediating between personal variables (e.g., empathy, adverse experiences, hostile rumination, trait cynicism, callous and unemotional traits, peer rejection, parent attachment, and positive parenting) and deviant and unethical behaviors [18,51,66,67,68,69,70,71,72,73].

Several empirical works support the role of empathy in reducing violent antisocial behavior through MD. Zych and Llorent [53] found in young people (16–22 years old) that affective empathy predicts the perpetration of bullying directly and through MD (partial mediation). Likewise, Wang et al. [25] found that MD partially mediated the reducing effect of one-dimensional empathy on aggression in young offenders (14–26 years). From a longitudinal perspective, Hyde et al. [69] found that empathy at age 12 reduced antisocial behavior at ages 16 and 17 through lower levels of MD at age 15.

In bystanders of cyberbullying, MD working as a mediator has also been shown. People with high levels of emotional or cognitive empathy tend to exhibit low levels of MD. Therefore, they are more likely to act as defenders of the victims and less inclined to act as *reinforcers* or *outsiders* [54]. Conversely, Paciello et al. [47] found that high levels of empathy could promote an altruistic response (propensity to help) by reducing the use of MD mechanisms, demonstrating that MD is a mediator of empathy for both antisocial and prosocial outcomes.

### 1.5. Measurement of Moral Disengagement

Several instruments have been created to measure MD, such as the Mechanisms of Moral Disengagement Scale (MMDS [16]), the Doping Moral Disengagement Scale [74], and the Propensity to Morally Disengage Scale [75]. Among them, the MMDS is the best known, used, and adapted due to the intercultural generalization of Bandura’s Social Cognitive Theory at the base of its development, demonstrating predictive capacity in studies from different countries, both Western (e.g., [23,68,76]) and Eastern (e.g., [18,25]).

The design and study of the psychometric properties of the original scale were carried out in Italy with students aged 10–15 years, resulting in a 32-item scale with a unifactorial structure [16]. However, the eight mechanisms of MD were presented as theoretical dimensions (of four items each) as well as the four MD strategies (cognitive restructuring, minimizing one’s agentive role, blaming or dehumanizing the victim, and disregarding or distorting the consequences) [6].

The original MMDS used a three-point Likert-type scale, which was also used by Pelton et al. [73]. However, Detert et al. [67] used a 5-option Likert scale in their psychometric study, confirming a structure of a second-order MD factor measured from 24 items grouped into the 8 first-order mechanisms and obtaining good internal consistency (*α* = 0.87). This response modality was replicated in subsequent studies, such as in the Danish [21], Spanish [23], and Mexican studies [55], although others have used Likert range scales of 4 [77] and 10 points [78].

In response to criticism about using measures with very general or broad items, other scales derived from the MMDS have been developed that consider contextual factors, since the same eight mechanisms described by Bandura can take different expressions depending on the context [79]. Thus, we can find studies using MD scales about school group dynamics to justify negative actions that are shared within a group (collective MD [80]), in the field of civic duties and interactions with organizations from the civil society (civil MD [79]), in the world of work [75], in a variety of legal contexts (e.g., [81]), or in sports contexts (e.g., [48]), among others.

The MMDS has been translated or adapted in adolescent and young people populations in several countries, showing good indicators of internal consistency but not always maintaining the unidimensionality of Bandura et al.’s scale [16]. In Spanish-speaking countries, there are adaptations of the original and uses of the Spanish version of the MMDS [30], both in the community population and in specific groups, with and without formal translation or adaptation processes in countries such as Mexico (e.g., [55]), Colombia (e.g., [61]), and Venezuela (e.g., [82]). However, despite contextual measures and adaptations in different countries, the MMDS by Bandura et al. [16] continues to be the most widely used scale to measure MD.

Table 1 reviews empirical studies published in journals indexed in the leading scientific databases (Web of Science, Scopus, and PsycInfo) that report the psychometric properties of MD scales in samples of adolescents and young people. The search included articles in English and Spanish without date restriction. The topic keywords were “moral disengagement” AND (adolescent* OR teen* OR juvenile OR “young people” OR youth OR “student*”) AND (psychometric properties OR validation OR scale OR questionnaire OR instrument). Some of the articles were included after reviewing the references of the articles selected in the formal strategy.

For all these reasons, and following the psychometric evidence that accounts for a mostly unifactorial structure, adequate internal consistency, and significant relationships with theoretically related constructs, added to the extension of its use in various countries and contexts, and also considering that it is a brief and easy-to-administer instrument, the MMDS is an excellent alternative to remedy the absence of instruments that measure MD in Chile. It is relevant to use brief risk assessment instruments by offering short, highly focused risk models to guide prevention and intervention programs [86]. This measure will help collect more precise information on MD and contribute to designing intervention plans to prevent and reduce the incidence and maintenance of pro-criminal cognitions that favor transgressive and criminal behavior in young people.

### 1.6. Objectives and Hypotheses

We aimed to (1) descriptively analyze the items of the MMDS, (2) demonstrate evidence of validity based on internal structure, (3) show evidence of reliability by internal consistency, and (4) demonstrate evidence of the validity of the MMDS based on its relationship with other theoretically linked antisocial and prosocial constructs.

The hypotheses to test are the following: (H1) The factorial structure of the MMDS will be one-dimensional, consistent with most studies; (H2) the internal consistency will be equal to or greater than 0.7; (H3) the total MD score will correlate positively with the scales (H3a) of abusive behavior against peers, (H3b) violent antisocial behavior, and (H3c) self-reported delinquency and (H4) negatively with (H4a) prosocial behavior and (H4b) empathy; (H5) when comparing low and highly empathic people in their MD means, a large effect size is obtained; and (H6) moral disengagement mediates the negative relationship between empathy and violent antisocial behavior.

## 2. Materials and Methods

### 2.1. Design

This work corresponds to an instrumental study [87] since it evaluates the psychometric properties of an instrument. On the other hand, we considered the methodological recommendations of Abad et al. [88] regarding the evidence of validity and reliability and the selection of statistical analyses to demonstrate them.

### 2.2. Participants

The sample consisted of 528 adolescents and was obtained through intentional non-probabilistic sampling. The inclusion criteria were the following: (1) age (between 14 and 18 years) and (2) being enrolled in an educational institution of the Chilean school system. In addition, it was sought to balance this with the following: (1) sex (48.9% men and 51.1% women) [89], (2) proportional population distribution according to socioeconomic stratification in Chile (groups in ascending order, where the E group corresponds to the poorest people and AB to the wealthiest (14% *E*, 35.9% *D*, 24.7% *C3*, 11.2% *C2*, 6.3% *C1b*, 6% *C1a*, and 1.8% *AB*) [90], and (3) population density by the geographic macro-zone of the country (13% north, 73% center, and 14% south) [89]. Given that the participation was voluntary in the online survey, the expected percentages of proportionality were not achieved exactly, although all the planned subgroups were represented in these three variables.

The adolescents surveyed were mostly women (62.5%), aged 17 and 18 years (72.5%), of low socioeconomic status (groups E and D: 61.1%), and residing in the central zone (68.8%) of the country (see Table 2).

### 2.3. Instruments

#### 2.3.1. Mechanisms of Moral Disengagement Scale (MMDS)

The Mechanisms of Moral Disengagement Scale, originally developed by Bandura et al. [16], is a self-reporting tool with 32 items (4 for each mechanism). The response format is a Likert type with five options (1= *strongly disagree*, 2 = *disagree*, 3 = *undecided*, 4 = *agree*, and 5 = *strongly agree*), and all statements are directly phrased. An example of an item is “*It is alright to fight to protect your friends*”. The original study reported adequate reliability (α = 0.82) and a unifactorial structure with no evidence of the appearance of empirical sub-factor [16].

To use the MMDS, we obtained permission from Albert Bandura, the copyright holder. Then, the scale was translated and back-translated to ensure equivalence between the original and the translated version in terms of language, level of abstraction, format and length of each item, grammatical structures, and scoring formats [91]. Two native English speakers with a certified domain of Spanish were translators. Subsequently, three experts in legal psychology, fluent in both languages and experienced in intervening with adolescents, were consulted. They reviewed the translation of the items and evaluated their conceptual and linguistic relevance according to the study population [91]. Finally, an online pilot survey was carried out with 73 undergraduate students. The participants were asked if any statement or word was doubtful or confusing and about the response time and order of the scale. Adjustments were made with this information. Appendix A contains the items of the Chilean version of the scale. Since the MMDS is the focal instrument of the study, its psychometric properties will be presented in the results section.

#### 2.3.2. Adolescent Social Behavior Self-Report Questionnaire (CACSA)

The Adolescent Social Behavior Self-Report Questionnaire (CACSA, due to its Spanish acronym) was developed by Alarcón et al. [92] and reviewed by Alarcón [93] to explore prosocial and antisocial behaviors by recording the frequency of behaviors expressed on a Likert scale from 1 (never) to 5 (always). The scale has 56 items grouped into 5 dimensions. In the validation study, the reliability of the scales ranged from 0.83 to 0.97, and evidence of validity based on the internal structure and relationship with theoretically related variables in Chilean adolescents was favorably demonstrated. For this study, the following scales were used: prosocial behavior (PROB; e.g., “*I have openly defended a classmate in a fight or argument*”), violent antisocial behavior (VAB; e.g., “*I have retaliated or taken out my anger by hitting others*”), and abusive behavior against peers (ABP; e.g.*,* “*I had fun during recesses at school taking snacks or money away from the little ones*”). In addition, we took the items of self-reported delinquency (SRD; e.g., “*I have taken clothes from a big store without paying for them*.”). With the data of this study, adequate internal consistency and a good fit of the factorial models to the data were demonstrated (see Table 3). These subscales were used to determine the evidence of concurrent validity.

#### 2.3.3. Toronto Empathy Questionnaire (TEQ)

The instrument was adapted to Chile [94] from the Toronto Empathy Questionnaire (TEQ) [95]. The TEQ stands out as a brief application instrument that allows the one-dimensional evaluation of adolescent empathy. It comprises 15 items with a Likert-type response format from 1 (never) to 5 (always). It establishes a general appreciation of empathy in its mainly affective component, preserving the characteristics of the cognitive dimension. Its items collect the contents of emotional contagion, emotional understanding, sympathetic physiological arousal, and altruism. An example of an item is “*When someone feels excited, I tend to get excited too*”.

In Chilean adolescents, the TEQ showed adequate internal consistency (*α* = 0.803), convergent validity evidence with the Cognitive and Affective Empathy Test (TECA), and discriminative capacity between adolescents with high prosocial behavior and those with high antisocial behavior [94]. In the sample of this study, good internal consistency and good fit of a unidimensional model of nine items were obtained after eliminating six items with low factor loadings (see Table 3). The empathy summated score was used to demonstrate evidence of concurrent validity.

### 2.4. Procedure

Regarding data collection, we obtained the study sample through an online survey. The process was administered by two polling companies, Netquest and Offerwise, who collect data for social and market research through online panels. They have an extensive registry of voluntary enrollees who are potential study participants. The participants received an invitation message via email and entered the online questionnaire after logging into the survey platform. The companies guarantee that the person who logs in to answer the survey is the one who is, in fact, registered in the panel.

The average response time was 38 min. International market research guidelines ruled the data collection through the ISO 26362:2009 standard on evaluation criteria for panel suppliers and the panels themselves.

The Universidad de La Frontera Scientific Ethics Committee approved the informed consent and informed assent documents. All adolescents had the authorization of a responsible adult, who also virtually consented to their children’s participation. Anonymity, confidentiality, and the rights to withdraw and to know the global results of the study were guaranteed.

### 2.5. Data Analyses

First, item analysis was performed to extract information on their quality [88]. To demonstrate evidence of validity based on the internal structure, we used a cross-validation strategy, dividing the sample into two sub-samples of 264 participants each. We ran an Exploratory Factor Analysis (EFA) with the first one using a polychoric correlation matrix. With sub-sample 2, we carried out a Confirmatory Factor Analysis (CFA) using the robust estimator for categorical items that were unweighted least squares means and variance adjusted (ULSMV). There were no missing data.

To interpret the fit of the models to the data, we used root mean square error of approximation (RMSEA), Comparative Fit Index (CFI), Tucker–Lewis Index (TLI), standardized root mean square residual (SRMR), and the ratio between the chi-square and the degrees of freedom (*χ^2^/df*). A good fit was considered if *RMSEA* < 0.05, and a reasonable fit was considered if *RMSEA* < 0.08 [96]. On the other hand, *CFI* and *TLI* > 0.95 indicated a good fit, and *CFI* and *TLI* ≥ 0.90 designated a reasonable fit [97]. For the *SRMR*, a value less than 0.08 was adequate [97], and for the ratio *χ*^2^*/df*, a value less than 2 was considered a good fit [98]. Additionally, two convergent validity analyses of the items were performed: average variance extracted (*AVE*) and construct reliability (*CR*).

The internal consistency of the MMDS items was calculated using McDonald’s omega and ordinal alpha on a polychoric matrix, where values above 0.7 were considered acceptable [99]. To determine evidence of validity based on the relationship with other constructs, we used Spearman’s non-parametric coefficient and Welch’s Student’s t-test, since the parametric assumptions were not met [100]. The effect size was considered large when *d* > 0.8 [101]. We used structural equation modeling (SEM) to test the mediation hypothesis, where empathy decreases VAB through MD. We included gender, SES, and age as control variables. The analyses were executed using JASP 0.16 and Mplus 8.5.

## 3. Results

### 3.1. Item Analysis

For each item, the response categories’ percentages, means, standard deviations, skewness, and kurtosis were calculated (see Table 4). On average, the item from the full scale with the greatest support was “*1. It’s alright to fight to protect your friends*”, which belongs to the *moral justification* mechanism. On the contrary, the one that generated the most disagreement was “*24. Kids who get mistreated usually do things that deserve it*” from *attribution of blame*.

Regarding the form of the distribution of the answers, a predominance of positive skewness was observed, which means that, in general, the adolescents tended to not support the MD statements that the items collected. No item showed extreme values of skewness (greater than two in absolute value) nor of kurtosis (greater than seven in absolute value) [88].

On the other hand, in Table 5, information on the discrimination indices (corrected item-total correlation), factor loadings of a one-dimensional model with 32 items, and validity indices for each item (correlation with 5 theoretically related external criteria) are presented.

### 3.2. Evidence of Validity Based on the Internal Structure

First, a CFA was executed using the unidimensional model of 32 items based on Bandura’s theoretical scheme. The model did not fit satisfactorily (*χ^2^_(1182.419)_/df_(464)_* = 2.54; *RMSEA* = 0.077, *90% CI* [0.071, 0.082]; *CFI* = 0.879; *TLI* = 0.871). Alternatively, a series of models with factorial structures from other validation processes were verified, such as by testing a general MD factor with eight first-order factors, consistent with the MD mechanisms [55]. The fit was not reasonable (*χ^2^_(2,232.298)_/df_(458)_* = 4.87; *RMSEA* = 0.086, *90% CI* [0.082, 0.089]; *CFI* = 0.844; *TLI* = 0.831). The same occurred with the strategy of eliminating the item with the lowest factor loading of each mechanism in a 24-item, 1-factor model [67,78].

As none of the tested options provided a good fit, we decided to switch to an empirically guided strategy to select the best items, using the information of the complete sample (*n* = 528) and taking as exigency criteria to keep an item in the model if (1) the corrected item-total correlation was greater than 0.4 [102], (2) the factor loadings were greater than 0.4 [103], and (3) the correlation of the item with the related constructs (validity index of the item) was significant and at least of a small intensity, either positive or negative (*r* > |0.1|; [101]). There were seven criteria (see Table 5), and items were gradually eliminated.

First, the seven items that did not meet three or more criteria were eliminated, and we ran an EFA. As it was impossible to reconcile the model’s fit with the theoretical interpretability of the resulting factors, the number of items was progressively reduced, being increasingly strict with the fulfillment of the criteria in each step and testing each model after gradually eliminating items. The goal was to achieve a parsimonious, interpretable, and well-fitting model. We stopped after discarding 22 items, keeping the 10 that met the seven criteria. They grouped unidimensionally in the EFA (*χ^2^_(132.643)_/df_(35)_* = 3.79; *RMSEA* = 0.073, 90% *CI* [0.060, 0.086], *CFI* = 0.960, *TLI* = 0.949, *SRMR* = 0.040). Once we determined the one-factor solution, a cross-validation strategy was carried out to study the model’s stability [104].

The EFA of the 10 items loading on a single MD factor obtained an adequate fit (*χ^2^_(85.466)_/df_(35)_* = 2.40; *RMSEA* = 0.074, *90*% *CI* [0.054, 0.094]; *CFI* = 0.962; *TLI* = 0.951; *SRMR* = 0.054) with sample 1. The factor loadings ranged between 0.501 and 0.683. The CFA performed with sample 2 also provided acceptable fit results (*χ^2^_(104.622)_/df_(35)_* = 2.98; *RMSEA* = 0.088, 90% *CI* [0.069, 0.107]; *CFI* = 0.935; *TLI* = 0.916). The range of factor loadings was from 0.504 to 0.762. From both factor analyses, the average variance extracted (AVE) and the reliability of the construct (CR) were calculated. The AVEs were less than 0.5 (below the cut-off point to be acceptable), and the CRs were greater than 0.7, which represents an adequate reliability value [103] (see Table 6). The results support Hypothesis 1 about unidimensionality.

### 3.3. Internal Consistency

In sample 1, we obtained *Ω* = 0.862 and *α*_ordinal_ = 0.861, while in sample 2, we found *Ω* = 0.865 and *α*_ordinal_ = 0.863. Both coefficients accounted for good internal consistency, which supports Hypothesis 2 on the adequate reliability levels of the MMDS-10.

### 3.4. Evidence of Validity Based on the Relationships with Other Variables

Demonstrating evidence of concurrent validity, the MMDS-10 score correlated positively (*p*s < 0.001) with abusive behavior against peers (*ρ* = 0.250), violent antisocial behavior (*ρ* = 0.366), and self-reported delinquency (*ρ* = 0.380). These results support Hypothesis 3. Likewise, the correlations between MD and prosocial behavior and empathy were negative (*ρ =* −0.309 and *ρ =* −0.352, respectively), giving empirical support to Hypothesis 4. Table 7 contains these correlations.

To determine the evidence of discriminant validity, we used an extreme group comparison strategy. With the total empathy score, we created a categorical variable, selecting quartile 1 (*n* = 126; *M*_empathy_ = 47.4; *SD*_Empathy_ = 4.10) and quartile 4 (*n* = 106; *M*_empathy_ = 67.9; *SD*_Empathy_ = 2.45). Then, people with low (quartile 1) and high (quartile 4) levels of empathy were compared by their MD means. We obtained significant differences (*p* < 0.001) and a large effect size, confirming Hypothesis 5 (see Table 8).

To evaluate the direct effect of MD on VAB, and to demonstrate the mediation hypothesis of MD, we used SEM. First, we evaluated the direct effect of empathy on VAB in the absence of MD as a mediator (model 1). The model fit adequately (*χ^2^_(66.936)_/df_(32)_* = 2.09; *RMSEA* = 0.045, *90% CI* [0.030, 0.061]; *CFI* = 0.974; *TLI* = 0.964), and it was possible to demonstrate that this relationship was negative and significant (*B* = −0.253; *β* = −0.232; *SE* = 0.053; *p* < 0.001). The percentage of explained variance was 12.5%.

Then, we entered MD as a mediator in the relationship (model 2). This model obtained an adequate fit (*χ^2^_(283.472)_/df_(148)_* = 1.92; *RMSEA* = 0.045, *90% CI* [0.037, 0.053]; *CFI* = 0.953; *TLI* = 0.946) and explained 30.5% of the variance in VAB. The direct effect of empathy on VAB (in the presence of MD) was not significant (*B* = −0.012; *β* = −0.089; *SE* = 0.056; *p* = 0.111), but its direct effect on MD was, as expected, significant and negative (*B* = −0.031; *β* = −0.318; *SE* = 0.046; *p* < 0.001). Additionally, as anticipated, the direct effect of MD on VAB was positive and significant (*B* = 0.619; *β* = 0.459; *SE* = 0.059; *p* < 0.001). The indirect effect of empathy on VAB through MD was significant and negative (*B* = −0.019; *β* = −0.146; *SE* = 0.028; *p* < 0.001). Since the influence of empathy was only manifested through MD, it was concluded that there was total mediation. In both model 1 and model 2, the differences by gender, socioeconomic level, and age were controlled, with only gender being significant. This result supports Hypothesis 6, since it was shown that MD increased VAB and that empathy reduced VAB by reducing MD, a mechanism that explained more variance of the dependent variable. Figure 1 shows both models.

## 4. Discussion

MD is a set of cognitive mechanisms that deactivate moral self-regulation processes and thus help explain why people behave unethically without apparent guilt or self-censure [105]. Knowing how youths justify immoral behavior is important because moral justification and agency develop in adolescence [10], and if these processes are not resolved successfully, they can lead to the validation of transgressive behavior. MD is a risk factor for juvenile delinquency and its severity [43] and also for aggression and cyberbullying [42,44,45,46]. For this reason, it is relevant to have psychometric evidence for its measurement in Chile, where scales have not been created or adapted.

This study aimed to analyze the psychometric properties of the MMDS, looking for evidence of validity based on the factor structure, internal consistency, and the relationship with other constructs (concurrent and discriminant). The results confirm all the hypotheses, supporting a one-factor structure of the MMDS-10, had an internal consistency greater than 0.7, positive and negative relations with the related constructs, and predictive and mediating capacities of MD as an explanatory process of violent behavior in adolescents.

The descriptive results of the 32 items (objective 1) allowed us to identify those that generated greater and lesser agreement. Item 1 (“*It is alright to fight to protect your friends*”) (moral justification) was supported the most, which can be attributed to the importance of the peer group (in-group) in adolescence and the development of collective MD, as getting into a fight to protect or defend a friend may be an acceptable group norm, justifying violent defense against outgroup members [106,107]. On the contrary, item 18 (“*Taking someone’s bicycle without their permission is just “borrowing it*”) (euphemistic labeling) was accepted the least, which may be due to the fact that adolescents perceive that taking an object without someone’s permission is more serious because a legal norm is violated (theft), unlike other reprehensible behaviors of the MMDS such as lying, insulting, or hitting or mistreating someone, which do not violate laws [108].

In the MMDS-10, the behavior *locus* was represented with three items of *advantageous comparison* (3, 11, and 19) and three of *euphemistic labeling* (2, 10, and 18). At the victim *locus,* two *dehumanization* items remained (23 and 31). In the outcome *locus,* items 6 and 30 of *distortion of consequences* were kept. Considering the eliminated items, the MD construct did not share enough variance with any item of the agency *locus* (four items of *displacement of responsibility* and four of *diffusion of responsibility*) nor with some from the *locus* of the victim (four of *attribution of blame* and two of *dehumanization*), some from the behavior *locus* (four of *moral justification*, one of *advantageous comparison*, and one of *euphemistic labeling*), or two from the outcome *locus* (*distortion of consequences*).

The results of the second objective confirmed the hypothesis of unidimensional structure, consistent with psychometric studies in other cultural realities such as the United States [73], Denmark [21], Australia [28], Iran [78], or Spain [31]. However, the final composition differed from these works since we had to conduct an empirical reduction of items to obtain a parsimonious, interpretable scale with a good fit, which has not been replicated elsewhere.

Our article joins the majority of studies that have reduced the number of items when analyzing the scale’s internal structure. Only one of the studies [55] maintained the original composition: 32 items, a structure of 8 first-order factors (1 for each mechanism), and 1 second-order factor representing MD. In light of the results, we believe that the replication of the theoretical structure with the full scale seems to be a general difficulty, which was also confirmed in the Chilean adolescent population. The MD construct behaves differently depending on the population, and it is not always possible to demonstrate the eight factors that represent the theoretical mechanisms. Therefore, as with Azimpour et al. [78], we recommend studying the scale’s factor structure in specific populations of Chilean adolescents to verify its stability.

The third objective of the study sought to demonstrate that the items homogeneously measured MD, which was achieved with good internal consistency values, supporting the second hypothesis. The magnitudes were similar to those reported in another study where the number of items was reduced (0.83–0.85) [31]. In addition, the construct reliability (CR) results indicated internal convergence, which means that all the measures consistently represented the same latent construct [103]. Therefore, the final score of the MMDS-10 was reliable.

Concerning the fourth objective, the results of this study confirm the hypotheses proposed. Supporting hypothesis 3a, MD correlated positively with ABP, which is a result in line with the extensive literature that establishes its relationship with aggression in the adolescent population in general [42] and among peers at school in particular [34,35,36,37]. Thus, based on Bandura’s theory [6,14], the increase in abusive behavior in the participants is related to a greater deactivation of the processes that self-regulate behavior, preventing them from feeling self-condemnation in the face of the violation of moral norms and inhibiting transgressive behavior.

In addition, this positive relationship was replicated in the case of VAB and SRD, confirming Hypotheses 3b and 3c, and extending this interpretation to behaviors that violate, in addition to moral norms, legal norms [19,20,21,22]. Following what was pointed out by Bandura [14], MD emerges as an important cognitive basis to explain why some adolescents start offending and persist in it, so this result is consistent. The values of the positive correlations of MD with more serious behaviors such as VAB and SRD were of a medium intensity, being very similar to the average effect sizes of several meta-analyses [43,44]

On the other hand, and as stated in Hypothesis 4a, MD correlated significantly and negatively with PROB, which is consistent with the literature [7,13,16,47]. The participants with low levels of MD not only presented fewer transgressive behaviors but also carried out more prosocial behaviors. As an explanation, Bandura et al. [16] alluded to the relationship between the low activation of certain MD mechanisms and empathy, to which Paciello et al. [47] added the importance of low MD in moral responsibility toward other people in need as a stimulant in prosocial decision making.

Similarly, we verified Hypothesis 4b about the negative relationship between MD and empathy as well as Hypothesis H5 about the higher mean score in MD of the group of adolescents with low empathy. A possible explanation for this association and the differences in means is that the quality of early parenting contributes to the development of empathy, which in turn reduces the development of later MD strategies [69]. Compared with the correlation coefficients of Detert et al. [67] (*r* = −0.27) and that of Rubio-Garay et al. [23], (*r* = −0.21), our result (ρ = −0.35) showed a stronger association, which places empathy as an interesting protective factor for the use of MD strategies in Chilean adolescents. These positive and negative correlational results and the mean differences support the concurrent and discriminant validity of the MMDS-10’s total score.

With the structural equation model, we intended to confirm the bivariate relationships tested in H3 and H4 and, in addition, provide evidence of the mediating role of MD (H6), in line with Bandura's idea of process. This hypothesis was confirmed, since MD mediated the empathy–VAB relationship, increasing the percentage of explained variance from 12.5% (model 1) to 30.5% after including MD as a mediator (model 2). In other words, VAB is better explained in the presence of the MD. According to Schaefer and Bouwmeester [65], through MD, people reconstruct unethical behaviors, with the effect of deactivating self-sanctions and thus clearing the way for ethical transgressions. It should be noted that the direction and intensity of the direct and indirect coefficients were very similar to those found by Zych and Llorent [53]. They explained 36% of the variance of bullying perpetration with a similar model. However, our result differs from those of Wang et al. [25] and Zych and Llorent [53], since our mediation was total.

Regarding gender differences, our work joins the majority that indicates that men tend to score higher in MD [24,53], although it differs from the conclusions of the meta-analysis of Gini et al. [42], which point out that there are no statistically significant differences according to gender, even though the absolute levels of MD and aggressive behavior are usually higher in boys.

### 4.1. Implications

The psychometric evidence that we provided supports the measurement of the MD construct with the abbreviated MMDS. Férriz et al. [43] maintained that considering the MD when selecting risk assessment instruments with subjects who have committed crimes would improve the predictive and classification capacity of the evaluation. In addition, and according to the age–crime curve or decrease in offenses with age [109], MD tends to decline over time, so measuring it can help explain why offenders desist from crime instead of becoming criminals for life [14,17].

Given the relationship between MD and offending behaviors, specialized intervention programs could focus their work on cognitive change, considering the possibility of reducing the use of MD strategies has been demonstrated [110]. Evaluations and interventions may benefit from focusing not only on preventing the selective moral disengagement of moral standards but also on developing empathic concern and perspective-taking skills to prevent maladaptive outcomes [26]. Our results suggest that specific interventions promoting empathy and reducing MD can diminish violent antisocial behavior in adolescents [18]. At an educational level, it is relevant to include MD in broad prevention programs that seek to reduce violent interpersonal behavior and vandalism as well as other risk factors simultaneously in schools [111], which are privileged places for the promotion of prosocial behaviors and a usual context of violence during adolescence [42,45,46].

### 4.2. Limitations

Some limitations are necessary to consider when interpreting the results. The different sample sizes according to gender, where the ratio was three females to each male, show that male adolescents are underrepresented in the study. This result should encourage researchers to include more balanced samples. Data collected with a cross-sectional design prevent us from knowing whether the relationships among empathy, MD, and violent antisocial behavior are maintained over time or whether one variable is longitudinally influenced by others in the model. In addition, the self-reporting instruments are sensitive to response biases that may affect the scores’ reliability due to social desirability, so using a validated measure should help researchers to include social desirability as a control variable in future moral disengagement research. Despite the limitations mentioned above, this study has the strengths of having a large sample size and providing evidence of reliability and two forms of validity.

### 4.3. Future Research

Following Bandura [6], this work focused on the idea of MD as a process. For this reason, it was treated as a mediator. However, it is necessary to investigate the dynamic relationships between MD and the other constructs, considering a possible moderating role of MD. Some researchers consider it a relatively stable cognitive disposition or orientation, a topic addressed in works by Moore [64] or Wang et al. [25]. Some studies have found that MD can moderate the relationship between empathy and violent behavior [25,51] or that empathy moderates the relationship between MD and aggression [26]. Thus, more clarifying evidence is needed on the role of the MD.

Additionally, it is important to continue investigating the link between MD and victimization. Adolescents who are victims of bullying resort to cognitive restructuring to legitimize these behaviors as an acceptable alternative or normative trait in peer relationships, thus engaging in bullying behaviors in the future [8]. On the other hand, MD can also serve as a strategy to minimize the importance or damage of victimization and justify the aggressor’s behavior [8,112,113,114].

Considering a relational vision of violence, we recognize that several actors converge in it—bullies, victims, and bystanders (defenders, reinforcers, and outsiders)—so we suggest strengthening the evidence of the differential impact of MD on aggression according to the situational role that adolescents assume [54].

In the psychometric field, we call for the development of scales adapted to specific contexts, such as those that address group dynamics within schools, in line with the collective MD [80,115], or of transgressive citizen behavior in the field of civic MD, such as business and financial corruption, embezzlement, theft, vandalism, and damage to environmental and public properties [79].

It is necessary to continue studying the mediating role of MD, such as between early adversity and future maladaptive behaviors. Some studies suggest that MD should be considered a potential cognitive mechanism linking early risks and later deviant behaviors [69]. In addition, there is still little empirical evidence showing how moral disengagement is initiated [64]. One possible line of research is to consider early adverse experiences, which we just mentioned, as a predictor of the development of moral justifications for deviancy.

## 5. Conclusions

In conclusion, this study presents the first attempt to study the psychometric properties of a moral disengagement scale in Chilean adolescents and to provide evidence of the MD working as a self-regulating cognitive process, according to Bandura. Our results showed that a 10-item version of the scale could be used as a single, reliable summated score. We also demonstrated that moral disengagement positively influences violent behavior while it mediates the protective influence of empathy on violent behavior, and empathy inhibits the use of MD strategies. Therefore, adolescents demonstrate less violent antisocial behavior. This mechanism supports the notion of MD working as a self-regulatory process. Based on the results, we recommend using and interpreting the MMDS-10 score in Chilean adolescents to obtain useful information for prevention and intervention efforts.

We believe that reducing the use of moral disengagement strategies would also imply a decrease in the probability that adolescents engage in behaviors that harm others. We recommend developing programs to promote healthy coexistence climates and prevent young people from using cognitive processes that rationalize their transgressive actions, disconnecting them from prosocial moral standards.

## Figures and Tables

**Figure 1 ijerph-19-12249-f001:**
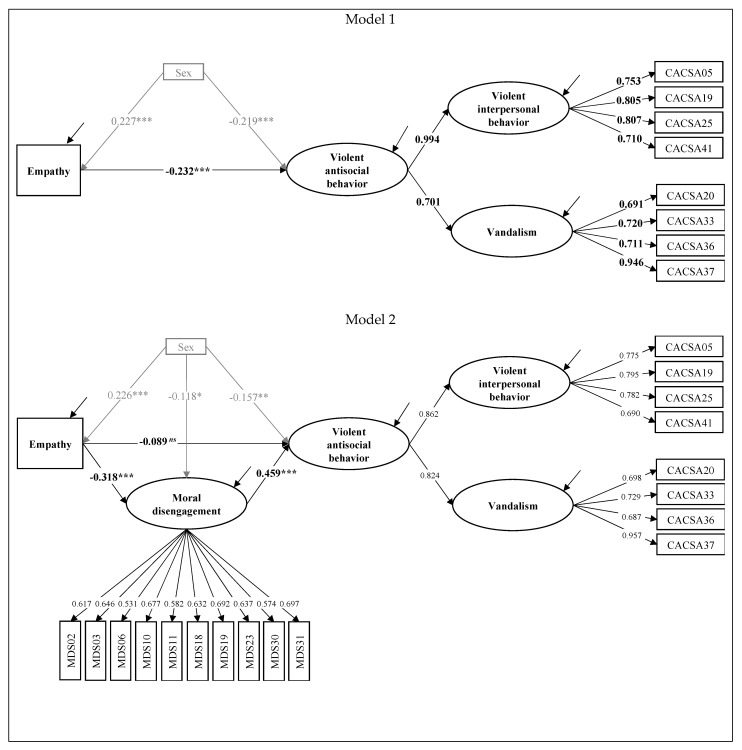
Predictive models of violent antisocial behavior without and with MD as a mediator. *Note*: **Model 1** = predictive model with control variable and without MD as a mediator; **Model 2** = predictive model with control variable and with MD as a mediator. *** = *p* < 0.001. ** = *p* < 0.01. * = *p* < 0.05. *ns* = not significant.

**Table 1 ijerph-19-12249-t001:** Psychometric properties of the MMDS in children, adolescents, and young people from different countries.

Country/Sample	Items and Factor structure	Reliability	Evidence of Validity Based on the Relationship with Other Variables
**United States** [73]			
245 children (mean age = 11.4 years)	28 items 1 dimension	α = 0.82	*Correlations with total MD (only significant):***self-ratings** (aggression, *r* = 0.30 **; delinquency, *r* = 0.26 **) and **parent ratings** (aggression, *r* = −0.13 *; social competence, *r* = −0.16 *). *Structural equation model*: positive effect of MD on aggressive behavior (*β* = 0.28 **) and delinquent behavior (*β* = 0.23 **).
**United States** [67]			
828 university students (mean age = 18.4 years)	24 items Eight first-order factors—one per mechanism—and one second-order factor (MD).	α _Total_ = 0.87 There is no information for each factor.	*Correlations with total MD (only significant):***sociodemographic** (gender (F= 0; M = 1) *r* = 0.30 **), **psychological** (empathy, *r* = −0.27 ***; trait cynicism, *r* = 0.30 ***; moral identity, *r* = −0.24 ***; unethical decisions, *r* = 0.34 ***), **and locus of control** (chance, *r* = 0.20 ***; power, *r* = 0.14 *). *Mediations*: (1) Moral disengagement mediates the relationships between empathy, trait cynicism, chance, locus of control, and moral identity and the dependent variable of unethical decisions (UD). (2) Moral disengagement does not mediate the relationships between internal locus of control and UD or between power, locus of control, and UD.
**Denmark** [21]			
677 students (11–14 years old)	32 items 1 dimension	α = 0.85	*Correlations with total MD (only significant)*: **peer-nominated** (bullying, *r* = 0.200 **) and **self-assessed** (victimization, *r* = 0.158 **; bullying, *r* = 0.280 **). *Group differences in MD*: **self-reported** **bullying status** (main effect (*η²* = 0.06 ***), *pure bullies* score higher) and **peer-nominated bullying status** (main effect (*η²* = 0.024 *), *bully-victim* status scored higher).
**Australia** [28]			
1022 students (12–15 years)	13 items 1 dimension	α = 0.90	*Correlations with total MD*: psychological distress, *r* = 0.231 **; resistive self-regulatory efficacy, *r* = −0.418 **; full serve of alcohol, *r* = 0.286 **; binged, *r* = 0.317 **; ever tried cannabis, *r* = 0.290 **.
**Mexico** [77]			
157 students (12–14 years old)	8 items Two factors: (1) moral justification and (2) language distortion and advantageous comparison.	α _F1_ = 0.69, α _F2_ = 0.69. α _Total_ = 0.73	Students labeled *pro-bullying* scored higher on *total MD* than *defenders* and *outsiders* (*p* < 0.001).
**Spain** [23]			
513 youths (15–25 years old)	32 items 8 indicators (mechanism) Three first-order factors: (1) depersonalization, (2) irresponsibility, and (3) rationalization, and a second-order factor (MD).	α _F1_ = 0.73, α _F2_ = 0.70, α _F3_ = 0.79 α _Total_ = 0.87	*Correlations with total MD (only significant):***aggression** (physical aggression: *r* = 0.55 **; verbal aggression, *r* = 0.41 **; hostility, *r* = 0.33 **; anger, *r* = 0.36 **) and **interpersonal reactivity** (perspective-taking, *r* = −0.30 **; fantasy, *r* = −0.11 *; empathic concern, *r* = −0.21 **.
**Mexico** [83]			
195 teenagers (ages not reported)	32 items There is no factorial structure study. The original structure is used.	α _Total_ = 0.80 There is no information for each factor.	*Correlations with total MD (only significant):* school violence, *r* = 0.37 *, victimization, *r* = −0.18 *, cultural beliefs (vertical individualism, *r* = 0.27 *) and attitudes toward violence (form of entertainment, *r* = 0.62 *; improve self-esteem, *r* = 0.53 *; managing problems and social relationships, *r* = 0.55 *; legitimate perception, *r* = 0.49 *). *Hierarchical regression*: among other factors, the MD predicted school violence (*β* = 0.40 *) and victimization experience (*β* = −0.28 *).
**Mexico** [55]			
1212 students (11–15 years old)	32 items Eight first-order factors, with one per mechanism. A second-order factor (MD).	McDonald’s Ω _Total_ = 0.93 There is no information for each factor.	*Structural Equation Model*: Positive effect of MD on encouraging aggression (*β* = 0.53 ***).
**Iran** [78]			
346 university students (average age = 21 years)	24 items One dimensional.	α of each dimension or mechanism ranging from 0.48 to 0.85. α _Total_ = 0.81 Stability, *r* = 0.693	*Correlations with total MD (only significant)*: social desirability, *r* = −0.124 *. *Group differences in MD (only significant)*: males scored higher (*η²* = 0.032 **).
**Spain** [31]			
1113 adolescents (12–17 years old)	14 items One dimensional.	α = 0.83	*Correlations with total MD*: sexism (hostile sexism, *r* = 0.46 ***; benevolent sexism, *r* = 0.41 ***) and dating violence (psychological aggression, *r* = 0.24 ***; physical aggression, *r* = 0.32 ***).
**Colombia** [84]			
827 students (11–16 years old)	32 items There is no factorial structure study. Original structure is used.	McDonald’s Ω = 0.94 There is no information for each factor.	*Direct effect*: adolescents who endorse higher levels of moral disengagement tend to be more aggressive (*β* = 0.09). *MD mediations*: Different predictors act on aggression through moral disengagement: distant maternal permissiveness (β _Total_ = 0.01), maternal trust (*β* _Total_ = 0.01), maternal verbal discipline (*β* _Total_ = 0.01), punishment-based discipline from the father (*β* _Total_ = 0.0001), the value of power (*β* _Total_ = −0.01), emotional instability (*β* _Total_ = 0.01), and energy (*β* _Total_ = 0.01).
**Spain and Colombia** [85]			
1396 Spanish and 1298 Colombian students (11–17 years old)	24 items 8 factors Measurement invariance was demonstrated.	Spanish: ρ*c* = 0.77–0.89 Colombian: ρ*c* = 0.84–0.91	*Correlations with the MD dimensions (Spanish or Colombian):***bullying perpetration** ((1) MJ, *r* = 0.36/.32; (2) EL, *r* = 0.49/.36; (3) AC, *r* = 0.42/0.30; (4) RD, *r* = 0.24/0.22; (5) DifR, *r* = 0.21/0.17; (6) CD, *r* = 0.42/0.32; (7) D, *r* = 0.32/0.24; (8) AB, *r* = 0.30/0.26) and **bullying victimization** ((1) MJ, *r* = 0.12/0.16; (2) EL, *r* = 0.14/0.20; (3) AC, *r* = 0.13/0.17; (4) RD, *r* = 0.12/.17; (5) DifR, *r* = 0.10/0.10; (6) CD, *r* = 0.10/0.14; (7) D, *r* = 0.13/0.17; (8) AB, *r* = 0.12/0.12). All correlations were significant at ***.

*Note:* MJ = moral justification; EL = euphemistic language; AC = advantageous comparison; DR = displacement of responsibility; DifR = diffusion of responsibility; DC = distorting consequences; D = dehumanization; AB = attribution of blame. * *p* < 0.05. ** *p* < 0.01. *** *p* < 0.001. ρ*c* = composite reliability.

**Table 2 ijerph-19-12249-t002:** Sample characteristics.

Socio-Demographic Variables		*f*	*%*	*χ2 (df)*	*p*
*Gender*	Male	198	37.5%	33.000 (1)	<0.001
	Female	330	62.5%		
*Age*	14	20	3.8%	326.754 (4)	<0.001
	15	34	6.4%		
	16	91	17.2%		
	17	132	25.0%		
	18	251	47.5%		
*SES*	E	139	26.3%	140.087 (4)	<0.001
	D	184	34.8%		
	C3	115	21.8%		
	C2	52	9.8%		
	C1a-C1b	38	7.2%		
*Country*	North	50	9.5%	125.894 (3)	<0.001
*Zone*	Central–Non-Metropolitan	133	25.2%		
	Central–Metropolitan (Santiago)	230	43.6%		
	South	115	21.8%		

*Note*: *SES* = socioeconomic status. E and D are typically classified as *low SES*, C3 and C2 as *Middle SES*, and C1a–C1b and AB as *High SES*. No AB SES participants were polled. *f* = frequencies; *χ*^2^ = one sample Chi-square test; *df* = degrees of freedom. All the frequencies of the categories within each variable presented statistically significant differences.

**Table 3 ijerph-19-12249-t003:** Evidence of reliability and fit of the factor structures of the constructs correlated with the MMDS.

Scales	α_ordinal_	Ω	χ^2^/df	CFI	TLI	RMSEA (90% CI)	SRMR
Abusive Behavior against Peers (ABP, 9 items) *Unifactorial*	0.876	0.860	3.94	0.948	0.925	0.075 (0.059 0.090)	0.062
Violent Antisocial Behavior (VAB, 8 items) *One general factor and two first-order factors (Violent Interpersonal Behavior and Vandalism)*	0.885	0.863	3.36	0.962	0.950	0.067 (0.050 0.084)	0.055
Self-Reported Delinquent Behavior (SRD, 9 items) *Unifactorial*	0.921	0.920	3.45	0.950	0.934	0.068 (0.053 0.084)	0.085
Prosocial Behavior (PROB, 10 items) *Unifactorial*	0.828	0.828	3.44	0.960	0.949	0.068 (0.055 0.081)	0.038
Empathy (9 items) *Unifactorial*	0.884	0.886	3.71	0.981	0.973	0.072 (0.057 0.087)	0.027

*Note*: χ^2^/df = Chi-square/degrees of freedom ratio; *CFI* = Comparative Fit Index; *TLI* = Tucker–Lewis Index; *RMSEA* = root mean square error of approximation; SRMR = standardized root mean square residual.

**Table 4 ijerph-19-12249-t004:** Percentages of the response options and descriptives of the MMDS items.

MMDS Items	SD	D	Und.	A	SA	M	SD	Skew.	Kurt.
1. It is alright to fight to protect your friends.	11.9%	14.4%	35.8%	28,0%	9.8%	3.1	1.135	−0.273	−0.583
**2. Slapping and shoving someone is just a way of joking.**	51.3%	31.3%	15.7%	1.7%	0,0%	1.7	0.798	0.846	−0.294
**3. Damaging some property is no big deal when you consider that others are beating people up.**	57.6%	27.3%	10,0%	3.8%	1.3%	1.6	0.905	1.512	1.981
4. A kid in a gang should not be blamed for the trouble the gang causes.	23.5%	22.7%	32.6%	16.7%	4.5%	2.6	1.151	0.166	−0.850
5. If kids are living under bad conditions, they cannot be blamed for behaving aggressively.	25.4%	31.6%	25.4%	13.4%	4.2%	2.4	1.126	0.454	−0.610
**6. It is okay to tell small lies because they don’t really do any harm.**	19.9%	30.5%	29.5%	17,0%	3,0%	2.5	1.083	0.219	−0.764
7. Some people deserve to be treated like animals.	44.5%	18.8%	19.9%	10.4%	6.4%	2.2	1.272	0.766	−0.575
8. If kids fight and misbehave in school, it is their teacher’s fault.	56.1%	30.9%	11.6%	1.3%	0.2%	1.6	0.759	1.138	0.766
9. It is alright to beat someone who bad mouths your family.	19.7%	26.1%	31.4%	17.6%	5.1%	2.6	1.136	0.177	−0.778
**10. To hit obnoxious classmates is just giving them “a lesson”.**	44.1%	28.2%	18.9%	6.8%	1.9%	1.9	1.036	0.900	0.037
**11. Stealing some money is not too serious compared to those who steal a lot of money.**	50.4%	30.9%	10.4%	5.5%	2.8%	1.8	1.022	1.372	1.371
12. A kid who only suggests breaking rules should not be blamed if other kids go ahead and do it.	39.8%	27.1%	20.6%	9.3%	3.2%	2.1	1.122	0.766	−0.306
13. If kids are not disciplined, they should not be blamed for misbehaving.	22.5%	23.1%	27.8%	21.6%	4.9%	2.6	1.189	0.108	−1.015
14. Children do not mind being teased because it shows interest in them.	47.9%	28.2%	20.5%	3.4%	0,0%	1.8	0.883	0.714	−0.626
15. It is okay to treat badly somebody who behaved like a “worm”.	26.1%	26.5%	25.2%	18.6%	3.6%	2.5	1.167	0.271	−0.970
16. If people are careless where they leave their things, it is their own fault if they get stolen.	43,0%	27.5%	12.5%	13.3%	3.8%	2.1	1.193	0.867	−0.383
17. It is alright to fight when your group’s honor is threatened.	34.8%	30.7%	21.8%	9.3%	3.4%	2.2	1.105	0.710	−0.281
**18. Taking someone’s bicycle without their permission is just “borrowing it”.**	66.3%	26.5%	5.3%	1.3%	0.6%	1.4	0.707	1.931	4.528
**19. It is okay to insult a classmate because beating him or her is worse.**	47.5%	25.2%	17.2%	8,0%	2.1%	1.9	1.072	0.961	0.012
20. If a group decides together to do something harmful, it is unfair to blame any kid in the group for it.	29.9%	19.5%	22.7%	18.4%	9.5%	2.6	1.335	0.283	−1.136
21. Kids cannot be blamed for using bad words when all their friends do it.	26.9%	28,0%	24.2%	14.8%	6.1%	2.5	1.203	0.431	−0.775
22. Teasing someone does not really hurt them.	65.5%	23.9%	8.5%	1.5%	0.6%	1.5	0.762	1.719	3.001
**23. Someone who is obnoxious does not deserve to be treated like a human being.**	41.3%	30.9%	18.2%	7.4%	2.3%	2.0	1.047	0.896	0.083
24. Kids who get mistreated usually do things that deserve it.	61.6%	25.6%	10.4%	2.1%	0.4%	1.5	0.793	1.446	1.702
25. It is alright to lie to keep your friends out of trouble.	19.7%	25.9%	33.9%	17.4%	3,0%	2.6	1.082	0.096	−0.785
26. It is not a bad thing to “get high” once in a while.	40.5%	18.6%	20.8%	14,0%	6.1%	2.3	1.286	0.582	−0.880
27. Compared to the illegal things people do, taking some things from a store without paying for them is not very serious.	53.2%	28,0%	11.4%	6.1%	1.3%	1.7	0.971	1.283	0.988
28. It is unfair to blame a child who had only a small part in the harm caused by a group.	22,0%	33.7%	27.1%	12.7%	4.5%	2.4	1.102	0.453	−0.487
29. Kids cannot be blamed for misbehaving if their friends pressured them to do it.	26.1%	29.5%	25.9%	14.4%	4,0%	2.4	1.137	0.402	−0.717
**30. Insults among children do not hurt anyone.**	40,0%	27.3%	23.5%	7.4%	1.9%	2.0	1.049	0.702	−0.332
**31. Some people have to be treated roughly because they lack feelings that can be hurt.**	48.7%	29.2%	17.2%	4,0%	0.9%	1.8	0.929	1.006	0.392
32. Children are not at fault for misbehaving if their parents force them too much.	23.3%	25.8%	32.6%	13.4%	4.9%	2.5	1.133	0.283	−0.674

*Note: SD* = strongly disagree; *D* = disagree; *Und.* = undecided; *A* = agree; *SA* = strongly agree; *M =* mean; *SD* = standard deviation. The items of the short version of the scale are in bold.

**Table 5 ijerph-19-12249-t005:** Discrimination indices (ITCc), factor loadings, and validity indices of the MMDS items.

MMDS Items	ITC_c_	Loading	ABP	VAB	SRD	PROB	Empathy
1. It is alright to fight to protect your friends.	0.358	0.373	**0.109 ***	**0.294 ****	**0.209 ****	0.033	0.020
**2. Slapping and shoving someone is just a way of joking.**	**0.480**	**0.514**	**0.118 ****	**0.246 ****	**0.247 ****	**−0.112 ***	**−0.209 ****
**3. Damaging some property is no big deal when you consider that others are beating people up.**	**0.464**	**0.490**	**0.128 ****	**0.264 ****	**0.234 ****	**−0.125 ****	**−0.178 ****
4. A kid in a gang should not be blamed for the trouble the gang causes.	0.395	**0.405**	0.040	0.088 *	**0.127 ****	−0.031	0.010
5. If kids are living under bad conditions, they cannot be blamed for behaving aggressively.	0.375	0.374	0.079	0.072	0.06	−0.008	0.098 *
**6. It is okay to tell small lies because they don’t really do any harm.**	**0.495**	**0.519**	**0.105 ***	**0.150 ****	**0.290 ****	**−0.110 ***	**−0.135 ****
7. Some people deserve to be treated like animals.	0.395	**0.425**	**0.119 ****	**0.170 ****	**0.152 ****	**−0.113 ****	**−0.121 ****
8. If kids fight and misbehave in school, it is their teacher’s fault.	0.319	0.345	0.089 *	**0.183 ****	**0.116 ***	**−0.150 ****	**−0.235 ****
9. It is alright to beat someone who bad mouths your family.	**0.497**	**0.528**	**0.166 ****	**0.303 ****	**0.206 ****	−0.068	**−0.133 ****
**10. To hit obnoxious classmates is just giving them “a lesson”.**	**0.555**	**0.594**	**0.118 ****	**0.316 ****	**0.169 ****	**−0.150 ****	**−0.226 ****
**11. Stealing some money is not too serious compared to those who steal a lot of money.**	**0.491**	**0.509**	**0.198 ****	**0.146 ****	**0.290 ****	**−0.135 ****	**−0.163 ****
12. A kid who only suggests breaking rules should not be blamed if other kids go ahead and do it.	**0.529**	**0.553**	**0.120 ****	**0.170 ****	**0.225 ****	**−0.107 ***	−0.094 *
13. If kids are not disciplined, they should not be blamed for misbehaving.	**0.443**	**0.452**	0.072	**0.102 ***	**0.116 ***	0.017	0.100 *
14. Children do not mind being teased because it shows interest in them.	**0.479**	**0.515**	0.091 *	**0.161 ****	**0.133 ****	**−0.145 ****	**−0.184 ****
15. It is okay to treat badly somebody who behaved like a “worm”.	**0.590**	**0.617**	**0.144 ****	**0.262 ****	**0.204 ****	−0.053	−0.050
16. If people are careless where they leave their things, it is their own fault if they get stolen.	0.308	0.334	0.094 *	0.082	**0.101 ***	**−0.118 ****	**−0.150 ****
17. It is alright to fight when your group’s honor is threatened.	**0.574**	**0.601**	**0.152 ****	**0.270 ****	**0.149 ****	−0.029	**−0.133 ****
**18. Taking someone’s bicycle without their permission is just “borrowing it”.**	**0.432**	**0.463**	**0.115 ****	**0.139 ****	**0.185 ****	**−0.111 ***	**−0.202 ****
**19. It is okay to insult a classmate because beating him or her is worse.**	**0.548**	**0.587**	**0.201 ****	**0.211 ****	**0.260 ****	**−0.147 ****	**−0.218 ****
20. If a group decides together to do something harmful, it is unfair to blame any kid in the group for it.	0.393	0.398	**0.136 ****	**0.116 ****	**0.178 ****	0.015	−0.060
21. Kids cannot be blamed for using bad words when all their friends do it.	**0.525**	**0.539**	**0.110 ***	**0.151 ****	**0.157 ****	**−0.101 ***	−0.090
22. Teasing someone does not really hurt them.	**0.428**	**0.472**	0.085 *	**0.193 ****	**0.167 ****	**−0.158 ****	**−0.303 ****
**23. Someone who is obnoxious does not deserve to be treated like a human being.**	**0.556**	**0.590**	**0.131 ****	**0.162 ****	**0.141 ****	**−0.141 ****	**−0.206 ****
24. Kids who get mistreated usually do things that deserve it.	0.390	**0.432**	0.046	**0.168 ****	0.076	**−0.135 ****	**−0.214 ****
25. It is alright to lie to keep your friends out of trouble.	**0.550**	**0.565**	**0.156 ****	**0.170 ****	**0.238 ****	−0.068	−0.030
26. It is not a bad thing to “get high” once in a while.	0.324	0.337	**0.104 ***	**0.155 ****	**0.261 ****	−0.068	0.010
27. Compared to the illegal things people do, taking some things from a store without paying for them is not very serious.	**0.515**	**0.538**	**0.181 ****	**0.180 ****	**0.291 ****	−0.083	**−0.163 ****
28. It is unfair to blame a child who had only a small part in the harm caused by a group.	**0.485**	**0.495**	**0.164 ****	**0.137 ****	**0.224 ****	−0.039	−0.010
29. Kids cannot be blamed for misbehaving if their friends pressured them to do it.	**0.512**	**0.526**	**0.136 ****	0.080	**0.145 ****	−0.021	0.010
**30. Insults among children do not hurt anyone.**	**0.513**	**0.545**	**0.103 ***	**0.202 ****	**0.210 ****	**−0.123 ****	**−0.203 ****
**31. Some people have to be treated roughly because they lack feelings that can be hurt.**	**0.518**	**0.559**	**0.156 ****	**0.180 ****	**0.219 ****	**−0.195 ****	**−0.262 ****
32. Children are not at fault for misbehaving if their parents force them too much.	**0.537**	**0.547**	**0.216 ****	**0.169 ****	**0.192 ****	0.036	0.072

*Note*: ITC_c_ = corrected item-total correlation (Discrimination Index); ABP = abusive behavior against peers; VAB = violent antisocial behavior; SRD = self-reported delinquency; PROB = prosocial behavior; validity index = correlation of each item with ABP, VAB, SRD, PROB, and empathy. In bold are the items that met the criteria: CIT > 0.4, Loading > 0.4, and r > |0.1|. * = *p* < 0.05. ** = *p* < 0.01.

**Table 6 ijerph-19-12249-t006:** Factor loadings and convergence analyses of the 10-item version of the MMDS.

MMDS Items	EFA (*n* = 264)	CFA (*n* = 264)
2. Slapping and shoving someone is just a way of joking.	0.658	0.584
3. Damaging some property is no big deal when you consider that others are beating people up.	0.613	0.637
6. It is okay to tell small lies because they don’t really do any harm.	0.501	0.561
10. To hit obnoxious classmates is just giving them “a lesson”.	0.683	0.642
11. Stealing some money is not too serious compared to those who steal a lot of money.	0.595	0.507
18. Taking someone’s bicycle without their permission is just “borrowing it”.	0.592	0.642
19. It is okay to insult a classmate because beating him or her is worse.	0.653	0.723
23. Someone who is obnoxious does not deserve to be treated like a human being.	0.664	0.662
30. Insults among children do not hurt anyone.	0.618	0.566
31. Some people have to be treated roughly because they lack feelings that can be hurt.	0.669	0.772
Average variance extracted (AVE)	0.393	0.402
Construct reliability (CR)	0.865	0.869

*Note:* Exploratory Factor Analysis (EFA) was performed with sample 1, and Confirmatory Factor Analysis (CFA) was performed with sample 2.

**Table 7 ijerph-19-12249-t007:** Spearman correlations between MD (summation of the 10 items) and related constructs.

	Moral Disengagement	ABP	VAB	SRD	PROB
Abusive Behavior against Peers	0.250 ***	1			
Violent Antisocial Behavior	0.366 ***	0.456 ***	1		
Self-Reported Delinquency	0.380 ***	0.590 ***	0.491 ***	1	
Prosocial Behavior	−0.309 ***	−0.053	−0.171 **	−0.152 *	1
Empathy	−0.352 ***	−0.087	−0.182 **	−0.154 *	0.576 ***

*Note*: * *p* < 0.05. ** *p* < 0.01. *** *p* < 0.001.

**Table 8 ijerph-19-12249-t008:** Comparison between high- and low-empathy groups by their moral disengagement means.

Group or Quartile	n	Min.	Max.	Mean	Median	SD	Welch’s t	df	p	d
Low empathy	126	10	39	21.24	21.5	6.738	6.754	218.67	<0.001	0.875
High empathy	106	10	33	16.24	15.0	4.467				

*Note*: Low empathy corresponds to quartile 1 (<= 53 points). High empathy corresponds to quartile 4 (65 or more points). *SD* = standard deviation; *df* = degrees of freedom.

## Data Availability

Not applicable.

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
