# Peer review of "Moral Disengagement as a Self-Regulatory Cognitive Process of Transgressions: Psychometric Evidence of the Bandura Scale in Chilean Adolescents"

_ijerph, 2022, doi:10.3390/ijerph191912249_

Round 1

Reviewer 1 Report

Dear Sir/Mam

Please find bellow the requested review regarding the manuscript. The article contains a lot of information on the issue; some of the information is useful but there are a lot of details that are confusing and difficult to follow. The topic is very interesting and use of sources is appropriate. In addition, it has too many tables that are not necessary.

The article contains a lot of useful information on the issue. It is quite clear what is already known about this topic and the research question is clearly outlined. The abstract is too brief and Discussion section involve too much information. There must be a balance in the manuscript

Specifically

Introduction

Introduction section doesn’t involve too much information, while Conclusion section is too long. There is an asymmetry in the manuscript. Table 1 is not necessary.

Research methods

Method is unclear. The authors must explain with more details. How were the participants recruited?

Results

Results are unclear and confusing. The authors must explain more targeted and briefly.

Tables with demographic and other information are necessary.

Positive: There are some strengths of the article that could have an impact in the field, such as the topic and its impact on the existed literature. The manuscript is approved after major changes.

Author Response

Thank you very much for your review. I am attaching the answers in a pdf file.

Reviewer 2 Report

The article is described in great detail and seems to exhaust the subject of moral disengagement among Chilean adolescents. There are a few things that should be improved:
- the introduction seems too broad;
- Table 1 - How were the described studies selected?
- 2.2 Participants - what percentage of the total population is the study population?
- please explain the indications of socioeconomic groups (AB-E)
- did the actual Likert scales differ in the minimum values (0 or 1)?
- has the minimum sample size been calculated? how much was it?
- l150 p.12 "the most diasgreement" - please reformulate, because it is currently unknown what disagreement it is about
- Table 2 is very unreadable
- why do the authors use the 90% CI?
- l 16 p.16 - as the authors defined 'appropiateness'
- please do not use the abbreviation in the caption of table 7
- looking at the results, it seems that the authors performed a regression, but it is not described in the methods
- figure 1 is not readable

Author Response

(The authors gave the same response as above.)

Round 2

Reviewer 1 Report

Revisions are accepted. Congrats on your work